# Improving Peach Fruit Quality Traits Using Deficit Irrigation Strategies in Southern Tunisia Arid Area

**DOI:** 10.3390/plants11131656

**Published:** 2022-06-23

**Authors:** Ines Toumi, Olfa Zarrouk, Mohamed Ghrab, Kamel Nagaz

**Affiliations:** 1Arid Regions Institute, University of Gabes, Route de Djorf, km 22.5, Medenine 4119, Tunisia; kamelnagaz@yahoo.com; 2Association SFCOLAB-Collaborative Laboratory for Digital Innovation in Agriculture, Rua Cândido dos Reis n° 1, Espaço SFCOLAB, 2560-312 Torres Vedras, Portugal; 3LEAF—Linking Landscape, Environment, Agriculture and Food Research Centre, Associated Laboratory TERRA, Instituto Superior de Agronomia, Universidade de Lisboa, 1649-004 Lisboa, Portugal; 4Olive Institute, University of Sfax, BP 1087, Sfax 3000, Tunisia; mghrab@gmail.com

**Keywords:** *Prunus persica* L., secondary metabolites, sugars, organic acids, mineral elements

## Abstract

The peach (*Prunus persica* L.) is one of Tunisia’s major commercial fruit crops and is considered one of the biggest water consumers of all crops. In warm and arid areas of southern Tunisia, irrigation is necessary to ensure orchard longevity and high yield and fruit quality. Nevertheless, under water-scarcity conditions and low water quality, water management should rely on efficient deficit irrigation strategies. In this study, sustained deficit irrigation (DI) and partial root-zone drying (PRD_50_) at 50% of crop evapotranspiration (ETc) were evaluated for their impact on the primary and secondary metabolites of the peach fruit of early cultivar Flordastar grown in the Tataouine region. A full irrigation (FI) treatment at 100%, etc., was used as a control treatment. Color, dry-matter content, firmness, organic acids, sugars, phenolic compounds, vitamin C, β-carotene and minerals were assessed on harvested mature fruits. Dry-matter content and firmness increased significantly under DI and PRD_50_ (13% and 15.5%). DI fruit had the highest soluble-solid content (SSC), reaching Brix values of 14.3°. Fruit sorbitol and sucrose contents were not affected by Di and PRD_50_. Higher glucose in fruit juice was observed in PRD_50_ (23%) and DI (21.5%) compared to FI, which had the highest malic acid content (33.5–37%). Quinic and citric acids decreased with DI and PRD_50_, while almost all individual phenolic compounds increased with deficit irrigation. Hydroxycinnamates and anthocyanins were significantly higher in fruits harvested from DI and PRD_50_ treatments. Proanthocyanidins (catechin and epicatechin) were only improved by DI, while flavone compounds and vitamin C were not affected by irrigation restrictions. β-carotene was higher in fruits yielded under FI (0.71 mg/100 g DM) than DI and PRD_50_ (0.21–0.43 mg/100 g DM). Macro- and micronutrients significantly increased in DI and PRD_50_ fruit. A significant difference between DI and PRD_50_ fruits was observed for Zn and Fe concentrations. This research highlights the positive impact of reduced irrigation on bioactive-fruit quality attributes and the suitability of PRD_50_ and DI as tools for irrigation management in arid areas of southern Tunisia, contributing to water-saving in orchards and the improvement of fruit commercial value.

## 1. Introduction

Scarce water resources in arid and warm areas are a limiting factor for agricultural development. Climate-change forecasts for the western Mediterranean basin predict increases in average temperatures and precipitation reductions [1]. Consequently, more frequent and severe drought periods during the fruit-growing season are expected [2], increasing the negative impacts on agriculture production and crop productivity [3].

Crop irrigation became widespread across the countries of the Mediterranean basin as a solution to sustaining local production and to cope with ongoing climate change. This has led to increased intensive farming production in the Mediterranean region [4], with ca. 30% of the cropland being irrigated, making the agriculture sector the largest consumer of freshwater [5] in one of the most water-scarce regions. The enhanced pressure on water resources in arid areas is being met by rising global concerns on the need to reduce water consumption by irrigated crops [6].

In this regard, improving water-use productivity is crucial in orchard management, and can be achieved with the adoption of efficient irrigation strategies based on water supply restrictions. Deficit irrigation emerged as a potential strategy to allow crops to withstand mild water stress with little-to-no decrease in yield [7]. Several deficit irrigation strategies have been proposed, ranging from the fixed-rate application of full crop evapotranspiration (ETc) homogeneously along the growing season (as in sustained deficit irrigation, DI), to supply irrigation at specific phenological stages, as is the case for regulated deficit irrigation (RDI) [8]. The understanding of hydraulic and chemical signals governing crop responses to watering greatly stimulated new irrigation strategies, such as partial root-zone drying (PRD) [9,10], which involves an alternate watering to each side of the plant root system. 

Most existing studies have focused the effects of deficit irrigation strategies on water-use efficiency and yield. Despite the fact that deficit irrigation induces different crop/fruit quality parameters (both nutritional and health-promoting), only a small number of studies have addressed this effect, and even fewer in desert- and arid-land orchards. 

Fruit nutritional traits are mainly a function of primary and secondary metabolites, including antioxidants such as carotenoids, flavonoids and phenolic compounds. However, literature explaining the metabolic background of the impact of specific irrigation practices, such as PRD, on fruit quality is limited [11]. Fruit quality traits are highly dependent on environmental factors and cultural practices [12]. Depending on the species, cultivar, geographic area or climate, deficit irrigation could have contrasting effects on fruit composition, improving, reducing or maintaining fruit quality traits [7]. Du et al. [13] showed that DI and PRD could be strategic in optimizing apple fruit traits. However, Chaves et al. [14] reported that only moderate water stress improved the quality of grape berries, whereas mild stress could be detrimental, particularly for flavonoid accumulation [7,15]. When it comes to grapevine irrigation strategies, no differences were observed in fruit composition and yield between the PRD and RDI [16], whilst Zarrouk et al. [7,15] showed vintage differences in the accumulation and biosynthesis of grape berry secondary metabolites between DI and RDI. In peach, phenolic compounds were highly increased by water stress in the Suncrest cultivar [17], and Rahmati et al. [18] reported a significant increase in fruit firmness under severe stress conditions in the Elberta cultivar.

Peach (*Prunus persica* L.) is one of the most important fruit tree species in Tunisia, with over 136,000 tons produced during the 2019 growing season. A large part of the production was exported [19], highlighting the economic importance of this crop for Tunisian economic balance. Traditionally, most peach production areas were located in the country’s northern regions. However, in the last two decades, southern Tunisian regions registered an increase in peach orchard installation, mainly due to the region’s high potential for using a diversity of new, very early-maturing cultivars. The use of low-chilling cultivars also led to the extension of orchard areas, contributing to the valorization of these marginal conditions. In addition, the climate and soil characteristics of the region, with high temperatures, full sunlight, low humidity and highly drained soils with coarse texture, offer great potential to reduce the incidence of fungi and nematode development and their spread. The Flordastar cultivar is the most cultivated in Tunisian southern dry areas, thanks to its high adaptability to warm winters and hot summers, and its attractive economic profitability in terms of production, quality and harvest timing (225 chilling hours). However, the commercial maturity of Flordastar fruits is usually achieved in the middle of the spring season, presenting superior size and good flavor and appearance. 

Nevertheless, natural water resources in arid and warm Tunisian areas are very scarce and dominated by high saline waters, with an electrical conductivity of more than 2 dS⋅m^−1^ [20]. Consequently, deficit irrigation-management strategies of peach orchards in these regions are considered a crucial practice to reduce water use while ensuring high fruit production and quality. 

Limited research is available on the effects of different irrigation strategies and amounts on peach fruit metabolites, particularly under arid and saline conditions. Hence, the aim of this study is to evaluate the impact of two deficit irrigation strategies (DI and PRD_50_), compared to full irrigation (FI), on the internal and external fruit quality—assessed by fruit size, fruit firmness, dry-matter contents, and primary and secondary metabolites—of an early-maturing peach cultivar in warm and dry areas of Tunisia irrigated with saline water.

## 2. Materials and Methods

### 2.1. Field Conditions and Plant Material

The study was carried out in 2016 on a 0.5 ha plot, at a commercial farm in El Gorthab, Tataouine, south-east Tunisia (32°11′47″ N, 290 m above sea level). The soil is sandy loam textured with very low organic matter (0.80%). The bulk density is 1.59 g cm^−3^. Climate is of Mediterranean arid type, with hot, dry summers and excessively low rainfall (average temperature: 22.5 °C, precipitation: <100 mm). Daily meteorological data (rainfall (mm day^−1^) and air temperature (°C)) were collected using a weather station (BWS 200, Campbell Scientific, Loughborough, UK), located 15 km away, coupled with an automatic data logger CR200X (Campbell Scientific, Loughborough, UK).

Six-year-old peach trees (*Prunus persica* L.) of the Flordastar cultivar grafted on Garnem (*P. persica* × *P. dulcis*) rootstock were used. Trees were goblet trained, spaced 5 m×5 m apart, and managed according to standard conditions for fertilization and pest and disease control. Trees were hand thinned in March (28 days after full bloom) and harvested during the first week of May (60 days after full bloom).

Trees were drip-irrigated with two lateral lines per tree row, using 4 emitters per plant. The irrigation water is characterized by a high salinity level (electrical conductivity (EC) = 3.17 dS⋅m^−1^). Irrigation was applied without leaching fraction because, in our soils (Sandy soil), irrigation water can readily flush salts out of the root zone.

Trees were subjected to three irrigation treatments for 4 successive growing seasons (2013 to 2016), during which agronomic parameters, such as vegetative and fruit growth, fruit load and yield, were monitored. Exhaustive internal fruit analyses, the subject of this current research, were conducted only in 2016. 

### 2.2. Irrigation Treatments

Crop evapotranspiration (ETc) was estimated by multiplying reference evapotranspiration (ETo), calculated with Penman–Monteith methodology, by the seasonal crop coefficient for peach (Appendix A) [21], and adjusted by shaded area [22].

Three irrigation treatments were applied: full irrigation treatment (FI) with irrigation at 100% of ETc; sustained deficit irrigation (DI), with continuous irrigation at 50% of ETc; and partial root-zone drying irrigation (PRD_50_)_,_ consisting of alternately irrigating one side of the tree at 50% of ETc. Changing of the irrigation side was carried out every 10 days.

### 2.3. Sample Preparation and Quality Traits

Sampling was carried out at commercial maturity and consisted of 20 fruits per tree and 5 trees per treatment (total of 100 fruits per treatment). After harvesting, fruits were transported to the laboratory at 4 °C, selected, and divided into sub-samples. Fruit diameters were measured in randomly selected samples for each replicate using an electronic digital caliper (Stainless Hardened, PERFECT QUALITY SOLUTIONS). Two opposite parts of each fruit (sun-exposed and shaded) were stored at −20 °C. A sub-sample of 10 fruits was pressed using a juice extractor (Evertek) and filtered. Dry matter (DM) was obtained using a gravimetric method, and values were expressed as percentages. Briefly, the fresh (FW) and oven-dry (DW) fruit weights (65 °C/48 h) provided were registered, and dry matter was calculated as: %DM = (DW/FW) × 100

Soluble-solid content (SSC) was determined in peach juice of 20 fruits per treatment using a hand refractometer (Pocket Refractometer PAL-1, ATAGO CO., LTD., Tokyo, Japan), and the results were expressed in °Brix. The titratable acidity was determined by titrating 10 mL of juice with 0.1 N NaOH to pH 8.1, and expressed in g/100 g. Fruit firmness (N) was evaluated using a hand penetrometer (CNR CT F&V, Andilog, Vitrolles, France) in two opposite positions on the peaches. CIEL*a*b* color measurements (Minolta chromameter CR-350, Japan) were conducted directly on the fruit skin around the equatorial region in three different positions (10 fruits per treatment). The color index (CI) was obtained via the following equation:$$\frac{1000\times a*}{L*\times b*}$$

The total phenolic content was determined in fruit samples (20 fruits per treatment) using the colorimetric Folin–Ciocalteu method [23]. The absorbance was read at 760 nm (UV/Visible spectrophotometer, Shimadzu-1600UV, Kyoto, Japan) and the results were expressed as mg Gallic Acid Equivalents (GAE) per 100 g DM. 

### 2.4. Metabolic Analyses 

#### 2.4.1. Sugars and Organic Acid Metabolites

For sugar and acid determination, 30 mL juice samples were centrifuged (15 min at 10,000 rpm), and the supernatant was filtered through a 0.45 µm nylon filter. 

Sugars were determined using HPLC (UFLC XR, Shimadzu, Kyoto, Japan) equipped with a NH_2_ column (Shimadzu, Shim-pack GIST NH2, Kyoto, Japan) (3 μm; 250 mm × 3.0 mm) operated at 40 °C, and a refraction index detector (RID10A, Shimadzu, Kyoto, Japan). Sample injection volume was 5 μL and the mobile phase was composed of 13% water and 87% acetonitrile set at a flow rate of 0.40 mL⋅min^−1^. The identification and quantification of sample sugar composition was achieved by comparison with sugar standard’s (sucrose, fructose, glucose and sorbitol) retention time and external standard calibration curves (125–2000 ppm), respectively.

Organic acids were measured using an ultra-fast liquid chromatography system under the following conditions: LC-20AD XR binary pump; Hi-plex H column with 7.7 mm × 300 mm dimension at 50 °C, detection: SPD M20A (Shimadzu, Kyoto, Japan) at 210 nm; and mobile phase with H_2_SO_4_ 0.1 M in H_2_O, flow rate 0.6 mL⋅min^−1^, injection volume 10 μL. The calculation of concentrations was based on the external standard method with concentrations 10–1000 ppm.

The concentrations of acids and sugars in the fruit juice were calculated using Shimadzu LabSolutions software ver.5.42 (Kyoto, Japan).

#### 2.4.2. Secondary Metabolites 

For phenolic identification/quantification, a 5 g sample (frozen full fruit tissue) was extracted using 10 mL of water/methanol (2:8, *v*/*v*) containing 2 mM NaF solution. The extracts were centrifuged at 11,500 rpm for 15 min, and the supernatants were filtered using a 0.45 μm filter and injected into the LC-MS (Shimadzu UFLC XR system, Kyoto, Japan), for the identification of individual phenolic compounds in Flordastar peach fruits. The LC-MS was coupled with an Inertsil ODS-4 C18 column (Shimadzu, Kyoto, Japan (150 mm × 3.0 mm, 3μm) at 40 °C. The flow rate was 0.5 mL⋅min^−1^, the mobile phases used were A and B composed of 15% water, 5% MeOH and 0.2% Acetic acid, and 50% Acetonitrile, 50% water and 0.2% Acetic acid, respectively. The injection volume was 20 μL. The compounds were detected between 200 and 609 nm and identified by comparison with retention time of the standards of different phenolic compounds. The concentrations were calculated using a Shimadzu LabSolutions software ver.5.42. Data were expressed as mg 100 g^−^^1^ DM.

Vitamin C was evaluated using the method described by Iglesias et al. [24]. A 3 g sample (frozen fruit) was mixed with an aqueous solution (3% metaphosphoric acid and 8% acetic acid) and incubated for 3 h (dark conditions). After centrifugation (9000 rpm × 15 min), the supernatant was micro-filtered and injected into the HPLC system. A Shimadzu UFLC XR HPLC System was used, consisting of a LC-20ADXR pump, a quadrupole detector model 2020, operated by Shimadzu LabSolutions software. A C18 column (150 mm × 3.0 mm, 3μm) from Aquasil was used. The mobile phase consisted of water, 10 mmol ammonium formate and 0.1% formic acid, and the flow rate was 0.4 mL⋅min^−1^. The detection was at 214 nm, and the injection volume was 20 µL. Vitamin C concentration was determined via an external calibration curve (0–50 mg L^−1^). 

β-carotene was assessed using HPLC following the procedure described by Thompson et al. [25]. After juice centrifugation (9000 rpm × 15 min), the supernatant was micro-filtered and injected into the HPLC system. A Shimadzu UFLC XR HPLC System was used, consisting of an LC-20ADXR pump, a UV/Vis detector model PDA, operated by Shimadzu LabSolutions software. A C18 column (150 mm × 3.0 mm, 3μm) was used. The mobile phase consisted of 20:80 ethanol:methanol. The flow rate was 0.5 mL⋅min^−1^. The detection was at 450 nm, and the injection volume was 40 µL. β-carotene concentration was determined via an external calibration curve (0–10 mg L^−1^). The concentration was calculated and expressed as mg 100 g^−1^. 

All standards used in HPLC or LC-MS techniques were from LGC and Sigma-Aldrich (Saint Louis, MO, USA).

#### 2.4.3. Mineral Content

Mineral content was determined as previously described by El Falleh et al. [26] with some modifications. Fruit samples (500 g) were dried (65 °C) until at a constant weight, grounded, and further ashed at 550 °C. After acid digestion, phosphorus (P) was determined spectrophotometrically (Secomam 1000, Secomam SA, Alès, France) as detailed by El Falleh et al. [26], while potassium (K), calcium (Ca), magnesium (Mg), sodium (Na), iron (Fe), manganese (Mn) and zinc (Zn) were determined via atomic absorption spectrometry (iCE3000, Thermo-Scientific, Waltham, MA, USA), after sample calcination. Concentrations were expressed as mg 100 g^−1^ DW.

### 2.5. Statistical Analysis

Data were analyzed by using SPSS Statistics for Windows (Version 25.0, IBM SPSS Statistics, Armonk, NY, USA). One-way analysis of variance (ANOVA) was used to test for statistically significant differences between means obtained for the treatments. Means and standard deviation (SD) were reported for all measured parameters. All graphs were constructed using GraphPad Prism software version 5.03. A Duncan’s multiple range test (*p* ≤ 0.05) was used to determine the significance level of the data. Principal component analysis (PCA) was performed to analyze all quality traits of Flordastar peach fruit produced under different irrigation treatments (DI, PRD_50_ and FI).

## 3. Results

### 3.1. Meteorological Conditions and Irrigation Applied 

Precipitation was irregularly distributed throughout 2016 (Figure 1). Although marked variations were observed between seasons and months, precipitation was mainly concentrated in the autumn and winter months. The rainiest months were December, September and November, with an accumulated precipitation of 16.4, 13.8 and 6.6 mm, respectively. This amount of precipitation water is highly favorable for carbohydrate reserve translocation to the trunk and roots, as they play an important role in early growth during the subsequent year. In deciduous trees, the initial growth of reproductive and vegetative organs depends on the mobilization of carbohydrates stored in the roots [27]. Likewise, rainfall during the winter, especially in December before the end of the dormancy period, can positively affect root growth and expansion. 

A significant quantity of salt was accumulated in the root zone during the dormancy period (data not shown). The winter rainfall contributed to salt leaching around the tree root zone, allowing adequate root development.

Summer was the driest period of the year, with very low precipitation (3.4 mm, from June to late August). The air temperature, shown in Figure 1, peaked in July and August. 

The reference evapotranspiration (ET_0_) and crop evapotranspiration (ETc) distribution along 2016 are shown in Figure 2. The annual ET_0_ and ETc values were 1596.5 and 1137.1 mm, respectively, with maximum values registered in July. Irrigation started at the bud-break stage (January, 18) with different amounts of water (Table 1). According to the trees’ phenological stages, the water amounts were variable but similar for the two deficit irrigation treatments (DI and PRD_50_). Water savings were around 50% for DI and PRD_50._

### 3.2. Physical Characteristics of Fruits 

Fruit size, expressed as the fruit diameter, was statistically higher in FI fruits (60 ± 2.7 mm) (Figure 3) when compared to DI and PRD_50_ fruits (55 ± 0.5 mm). 

The dry-matter content of Flordastar fruits was inversely proportional to the amount of irrigation water applied (Figure 4). The lowest DM values were found in the FI fruit samples (14.20%). The average DM in both deficit irrigation treatments (DI and PRD_50_) was similar, at 16.33% and 16.85%.

The peach fruit samples from deficit irrigation treatments (DI and PRD_50_) were also firmer (25 and 36%, respectively) than fruit samples harvested from FI treatment (Figure 5), but no significant differences were observed between DI and PRD_50_. 

The skin color parameters (L*, a*, b*) were unaffected by the stress conditions, as fruit samples from FI, DI and PRD_50_ irrigation treatments provided similar responses (Appendix A). 

### 3.3. Peach Fruit Sugars and Organic Acid Content

SSC was highest in peach fruit samples harvested from deficit irrigation treatments, compared with those harvested from FI treatment. However, no significant differences were found between PRD_50_ and DI treatments (Table 2). 

Different trends were observed when looking at individual sugar compound contents (Table 2). No significant differences were observed in sucrose content among irrigation treatments. However, glucose content was increased under DI and PRD_50_ when compared to FI (by 23% and 21%, respectively). Fructose was significantly increased by DI treatment, with a 15% increase compared to other irrigation treatments. The sorbitol content in Flordastar peach samples was not significantly different among treatments, but FI fruits accumulated less. Peach fruit samples’ total sugar content at harvest was not significantly different among the tested irrigation strategies.

In general, titratable acidity was highest in FI and lowest in PRD_50_ fruit samples, while DI showed an intermediary behavior (Figure 6). The acidity of FI fruit samples was 17 and 22% higher than DI and PRD_50_, respectively. 

The FI treatment significantly increased malic acid content in the peach fruit compared with DI and PRDI_50_ (by ca. 35% in both treatments) while significantly decreasing citric acid and quinic acid content (Figure 6).

### 3.4. Secondary Metabolites

All polyphenol compounds obtained in Flordastar fruit samples are shown in Table 3. In general, the total phenolic content (measured using the Folin method) was similar, and no significant differences were observed among the FI, DI and PRD_50_ treatments.

Gallic acid (GA), as a phenolic acid, was significantly influenced by the irrigation treatment: GA was two times higher in DI and PRD_50_ than in FI fruit samples (Table 3).

The hydroxycinnamate group was the most abundant in Flordastar fruit samples, with three identified compounds: chlorogenic acid (CHA), neochlorogenic acid (NCHA) and cryptochlorogenic acid (CCHA) (Table 3). The CCHA was the main bioactive compound determined in the present study, and was significantly higher in the deficit irrigation treatments (DI and PRD_50_) (Table 3). The less abundant NCHA and CHA were also significantly higher in fruit samples grown under DI and PRD_50_ when compared with FI (by 60% in both deficit irrigation treatments). 

In the case of flavan-3-ols content, two different compounds were identified: catechin (CAT) and epicatechin (EPICAT), with CAT being the most abundant in Flordastar fruit (Table 3). Both CAT and EPICAT were higher in DI than in the FI fruit samples, at 30% and 43%, respectively. The differences between FI and PRD_50_ were not significant.

Only one anthocyanin compound, cyandin-3-glucoside (C3G), was detected in all measured fruit samples from the three irrigation treatments (Table 3). C3G was two times-higher in DI and PRD_50_ than in the FI-treated fruit samples.

Two flavonol compounds were identified: quercetin-3-galactoside (Q3GAL) and quercetin-3-rutinoside (Q3R). While no significant differences were observed among irrigation treatments for Q3GAL, Q3R fruit content was improved by PRD_50_ treatment, followed by DI (Table 3).

Other bioactive compounds such as flavones and vitamin C were also identified in Flordastar fruits; however, no significant differences were observed among the fruit samples of each treatment (Table 3). In contrast, β-carotene was significantly higher (70%) in FI fruit samples than in DI samples. Differences in β-carotene between water-stressed fruits were also observed: PRD_50_ fruit samples had double the β-carotene content determined in DI samples.

### 3.5. Mineral Content 

The effects of the deficit irrigation strategies on the macro- and micronutrients of peach fruit samples are shown in Figure 7 (A: macronutrients and B: micronutrients). In general, all macro- and micronutrient fruit contents were significantly increased by the DI and PRD_50_ treatments when compared to FI. No differences were observed between DI and PRD_50_ fruit samples, except for iron (Fe) and zinc (Zn) content.

Potassium (K) was the major mineral compound found in the fruit samples, representing more than 64% of fruit mineral composition, followed by calcium (Ca), with 18–24% (depending on the irrigation treatment). Fe was the most abundant micronutrient found, varying between 0.33% and 0.60% for the FI and DI fruit samples, respectively. Zn increased in PRD_50_ samples, while Mn was higher in the fruit samples of both deficit irrigation treatments.

The impact of irrigation treatments (FI, DI and PRD_50_) on internal and external peach fruit quality was assessed via principal component analysis (PCA). Total variation was fully explained by two components (F1 = 82.3% and F2 = 17.7%) (Appendix A), highlighting a strong differentiation pattern considering the first component, and separating the deficit irrigation treatments from full irrigation. Although less significantly, DI and PRD_50_ were clearly separated along the secondary component. A heatmap of Pearson’s correlation coefficient with hierarchical clustering analysis was conducted to understand the relationship between fruit quality traits and irrigation treatments (Figure 8). This test provided evidence that most parameters were inversely correlated between FI and the deficit irrigations (DI and PRD_50_). The heat map results showed that FI treatment increased β-carotene, titratable acidity and sucrose, but decreased minerals, quinic and citric acids, sorbitol, glucose, fructose and the majority of phenolic compounds. The DI and PRD_50_ treatments were positively correlated with most secondary metabolites, glucose, citric acid and sorbitol.

## 4. Discussion

The effects of partial root-zone drying (PRD_50_) and sustained deficit irrigation (DI) on the quality attributes of Flordastar peach fruit were investigated during the 2016 growing season. The results showed that most quality parameters were increased in both of the tested deficit irrigation treatments (Figure 3, Figure 4, Figure 5, Figure 6 and Figure 7, Table 2). 

The two deficit irrigations (DI and PRD_50_) increased the fruit dry-matter content and fruit firmness as compared with the FI treatment (Figure 4 and Figure 5), which is in agreement with several previous reports [18,28,29]. A positive relationship between fruit DM and fruit firmness has been demonstrated in several fruit crops [30] such as tomato [31], kiwi fruit [32] and blueberries [33,34]. Increased fruit firmness resulting from deficit irrigation has been previously attributed to cell-wall remodeling, particularly pectin methylesterification and xyloglucan acetylation [35,36]. Fruit cell walls are pectin-rich, and calcium–pectin cross-links are a major factor in determining their physical and structural properties [37]. Our results suggest that increased firmness in fruit under water stress is related to fruit calcium content, which was highest in DI and PRD_50_ (Figure 7). Indeed, previous reports showed the beneficial effect of calcium-containing spray treatments in peach fruit, particularly on cell integrity and disease resistance [38]. Fruit firmness largely determines the transportability and shelf-life of fruits, reducing mechanical damage, susceptibility and fungal or bacterial infection [39,40,41]. Additionally, firmer fruits are also the most appreciated by consumers. Hence, our results point out the overall suitability of deficit irrigation as an agronomic management practice to increase fruit shelf-life and acceptability.

Sugars and organic acids are the major primary metabolites for fleshy fruits, and their concentrations contribute to establishing fruit flavor [42]. In the present study, SSC was significantly increased by DI and PRD_50_ (Table 2). This result is in agreement with the report by Adu et al. [43], which concluded, through an exhaustive meta-analysis of 43 different fruit crops subjected to different deficit irrigation regimes, that DI and PRD_50_ improve SSC by 4 to 5%. Similar results have been reported in late-season peach fruit [44] or grape berries [45]. Lechaudel et al. [46] also reported an increase in SSC in mango fruit, arguing that the increased fruit DM, due to water-stress conditions, explains the increases in glucose and fructose concentrations. As indicated above, Flordastar peach fruit had higher DM under the DI and PRD_50_ treatments when compared with FI, which is concomitant with the increases in glucose and fructose in deficit irrigation conditions (Table 3). In addition, previous data from citrus [47] and plum fruit [48] explained the SSC increase by the active accumulation of sugars combined with decreased water potential and an increased concentration of dissolved substances. However, Rahmati et al. [18] explained the increase in fructose and glucose levels in late-maturing peaches by the decrease in energy cost for fruit growth under drought conditions (decrease in fruit size), which resulted in a decline in the metabolization of these simple sugars in the glycolysis process. In contrast, sucrose, the main fruit-soluble sugar content-wise, was less sensitive to the water regime (Table 3). Among plants’ synthesized sugars, sucrose is the main form of carbon found in the phloem, even though in some species [49,50,51]—including *Prunus* species such as peach and apricot [49,52]—polyols such as sorbitol are translocated. As for sucrose, our data showed no significant differences in sorbitol among irrigation strategies. Sucrose and sorbitol results suggest that no assimilate sink starvation occurred during Flordastar fruit ripening under DI and PRD_50_ treatments. 

Deficit irrigation (DI and PRD_50_) significantly decreased the titratable acidity in fruit, but stress impacted the content of individual organic acids differently (Figure 6). DI and PRD_50_ treatments enhanced quinic and citric acids, while the predominant organic acid, malic acid, was highest in fruits grown under FI (Table 3). Shikimic acid concentration (quinic and citric acids) has been shown to decrease with leaf/fruit ratio increase, while greater assimilate supply is related to high malate [53]. In accordance, our previous study showed that Flordastar trees subjected to DI and PRD_50_ have a high leaf/fruit ratio [54]. Additionally, potassium has a key role at a cellular level, whereby different mechanisms allow K^+^ to affect the metabolism and the storage of organic acids. Acidity results from the ratio between free organic acids and organic acids neutralized by K^+^ in grape berries. In warmer climates, K^+^ ion accumulation increases during grape ripening, leading to an excessive neutralization of organic acids [55,56,57]. The high K content found in Flordastar fruits can be related to the high temperatures experienced in southern Tunisia, and may explain the results found in DI and PRD_50_-treated fruits concerning the different organic acids. Considering the high level of sweetness conferred by the presence of fructose in fruit [58], the importance of sorbitol and organic acids to peach aroma and taste [50], and sucrose levels in ripening signaling, our results (Table 2) show that both of the studied deficit irrigation treatments had a positive effect on the quality parameters of early-maturing peach fruit.

The results in several crop species have demonstrated the beneficial impact of deficit irrigation on fruit quality and nutritional and bioactive values [11]. The benefits are attributed to a positive effect on the biosynthesis and accumulation of secondary metabolites, phytochemical compounds with known functional properties beneficial to human health and fruits’ antioxidant systems. Lopez et al. [28] showed an increase in total polyphenol content under severe water-stress conditions in late-maturing peach fruit, contrasting with the present work, wherein no effect was attributed to irrigation treatment. Previous reports by Falagan et al. [59], in the same variety, showed similar results in fruit pulp, while in fruit peels, total phenols were increased by water stress. Plant water status and meteorological conditions could partially explain these discrepancies among the reported data. Late-maturing fruit could suffer from a long period of cumulative heat/water stress that increases the degradation of several phenolic compounds [7], while in early-maturing fruit, severe stress conditions, if present, are limited in time and could have little impact on fruit phenolic content. 

Anthocyanin compounds are the main contributors to fruit coloration and visual attractiveness for consumers. The red color of peach pericarp and flesh is mostly due to the accumulation of cyanidin-3-glucoside and cyanidin-3-rutinoside [60]. Although no differences were observed among treatments regarding the CIELab parameters or the color index, cyanidin-3-O-glucoside (C3G), the main anthocyanin compound in Flordastar fruit, was higher in fruits grown under DI and PRD_50_. This result is aligned with different reports of several fruit species showing an increase in anthocyanin content due to water-stress conditions [7,61]. Sucrose has been related to the activation of anthocyanin biosynthesis enzymes as a signaling molecule in peach fruit [62]. However, in our study, although sucrose content in fruits was higher in deficit irrigations, the differences between treatments were not statistically significant (Table 2). In grape berries, anthocyanin–sugar decoupling due to high temperature (above 35 °C) has been reported [7,63]. Corroborating this hypothesis, our meteorological data for the 2016 season showed a maximum temperature (above 40 °C) in April (Figure 1). Moreover, high values of glucose and fructose in fruits grown under DI and PRD_50_ treatments were observed, suggesting an increase in sucrose degradation in water-stressed treatments compared with FI. The role of phosphorus has been reported as being key to fruit yield and the modulation of secondary metabolites [64]. In the present study, P content was higher in water-stressed fruit, which might lead to secondary metabolite modulation (e.g., anthocyanin biosynthesis) by the deficit irrigation treatments.

Contrasting with anthocyanin, β-carotene was highly repressed by DI and PRD_50_ treatments (Table 3). β-carotene is an important compound of the organoleptic quality of fruit, both for its color attributes and its role as an aroma precursor, such as norisoprene and monoterpene, during fruit ripening [65]. High contents of β-carotene were reported earlier by [66] in the Flordastar cultivar. Carotenoid accumulation in fruit is inversely correlated to sunlight exposure [67]. Furthermore, β-carotene provides substrate precursors for abscisic acid (ABA) phytohormone biosynthesis. Several reports showed the role of ABA in mediating the adaptation of the plant to water stress and its concentration increases in fruit subjected to a deficit irrigation regime [7,68]. Altogether, we hypothesize that a low content of β-carotene in Flordastar peach fruit in DI and PRD_50_ treatments could be related to its shift as a substrate for ABA biosynthesis. Carotenoid content was also reported to be variety-, agronomy- and environment-dependent [69]; thereby, further work is needed to shed light on the role of water deficit in its biosynthetic pathway in peach fruit. 

Hydroxycinnamic acids were highly accumulated under deficit irrigation conditions, while flavan-3-ols were highest in DI treatments (Table 3). Both phenolic groups are known as powerful antioxidant agents. Chlorogenic acid (CHA) has been shown to be the main antioxidant compound in peach fruit, rather than catechins and vitamin C [70]. In addition, fruits with higher CHA are more resistant to biotic stressors such as pathogens that cause brown rot [71]. In tomato fruits, chlorogenic acid was also enhanced in two different varieties subjected to water stress [72]. It can be concluded that deficit irrigation management is closely associated with the quality of peach fruit. 

Mineral compounds were highly impacted by the DI and PRD_50_ regimes (Figure 7), contrasting with previous studies, showing increased mineral content in well-irrigated fruit trees, as water facilitates ion absorption by the roots [73]. Different soil conditions could explain our results. Since Flordastar soil is mainly sandy and highly filtrating, the high amount of water in the FI regime could induce mineral leaching at the root zone, explaining the lower mineral content in FI fruits [74]. 

Mineral nutrients have a key role in fruit yield and quality [75,76]. In recent years, many studies have shed light on the influence and the molecular mechanisms for mineral nutrition (N, P, K, Ca and Fe) in several Rosaceae crops (fruit development, quality), especially highlighting the roles of P, K and Fe in fruit quality [64]. In fact, different environmental factors are also responsible for the final fruit mineral content. A more comprehensive understanding of the uptake and translocation system in peach for mineral elements is required, particularly under water-scarcity conditions. Notwithstanding the physiological meaning and reasons for fruit mineral increase, both of the deficit irrigation treatments enhanced the mineral availability in fruit, and thereby, its nutritional and bioactive attributes. Our results underline the relevance of deficit irrigation strategies such as DI and PRD_50_ to improve internal fruit quality, due to the higher concentration of primary and secondary metabolites. 

## 5. Conclusions

Our results support the need for and importance of using deficit irrigation strategies, such as DI and PRD_50_, as tools for the efficient water management of peach orchards in arid areas such as South Tunisia, where water scarcity is increasing due to climate change. This study suggests that allowing some degree of water stress in early-maturing peach trees, at 50% of ETc, represents an effective technique for water-saving and improving fruit quality parameters and, thereby, farmers’ economic returns.

The study shows the effectiveness of both deficit irrigations in increasing the physical and chemical parameters of fruit, which improve fruit shelf-life, resistance to pathogens, fruit nutritional and bioactive traits (secondary metabolite content and mineral content) and commercial value. However, although deficit irrigation regimes have been applied in the Flordastar orchard for 4 years, data only reference the last year of the experiments. The monitoring of different production seasons will shed light on inter-season variability. In addition, the long-term impact of using deficit irrigation techniques with low-quality water on soil salinization, as well as tree vigor and longevity, needs to be monitored.

## Figures and Tables

**Figure 1 plants-11-01656-f001:**
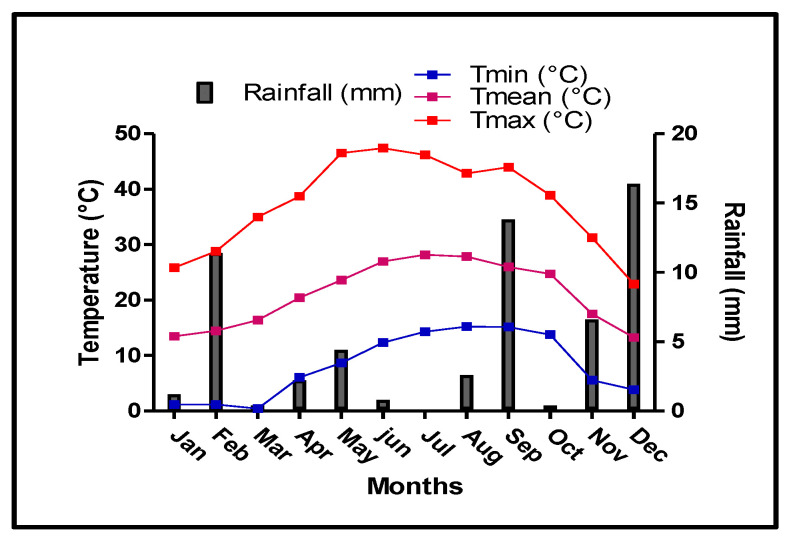
Monthly minimal, maximal and mean air temperature (Tmin, Tmax and Tmean, respectively) and rainfall during 2016.

**Figure 2 plants-11-01656-f002:**
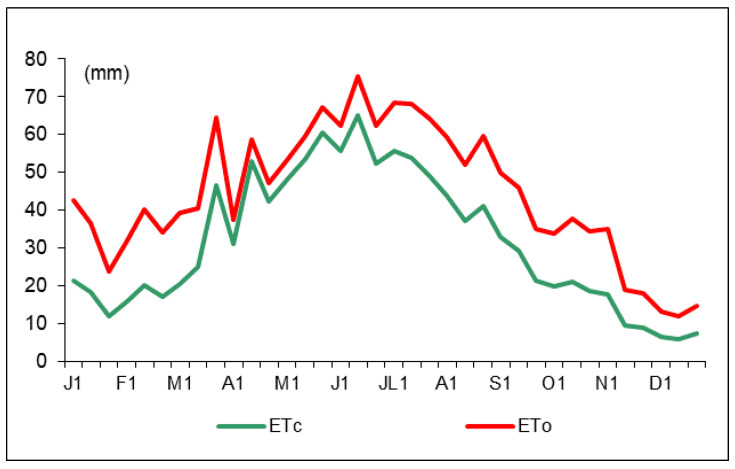
Ten-day reference evapotranspiration (ET_0_, red line) and crop evapotranspiration (ETc, green line) values during 2016.

**Figure 3 plants-11-01656-f003:**
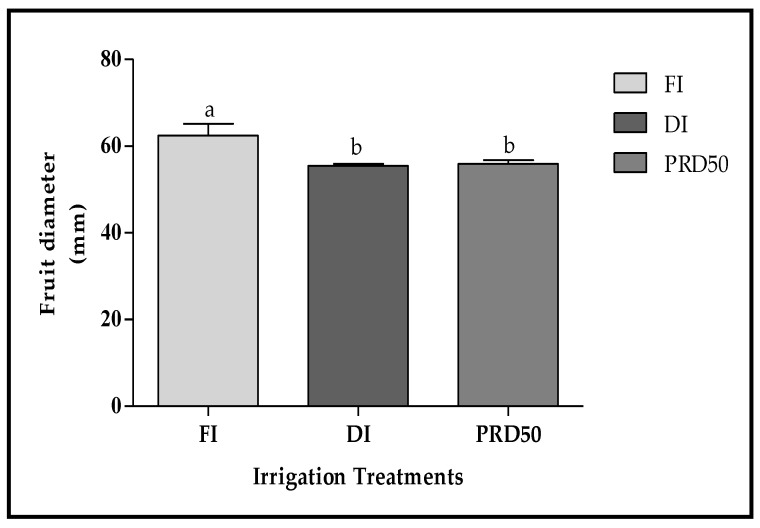
Fruit diameter (±SD) of Flordastar peaches grown under FI, DI and PRD_50_ irrigation treatments, expressed in (mm). Different letters refer to significant differences tested using Duncan’s multiple range test (*p* < 0.05).

**Figure 4 plants-11-01656-f004:**
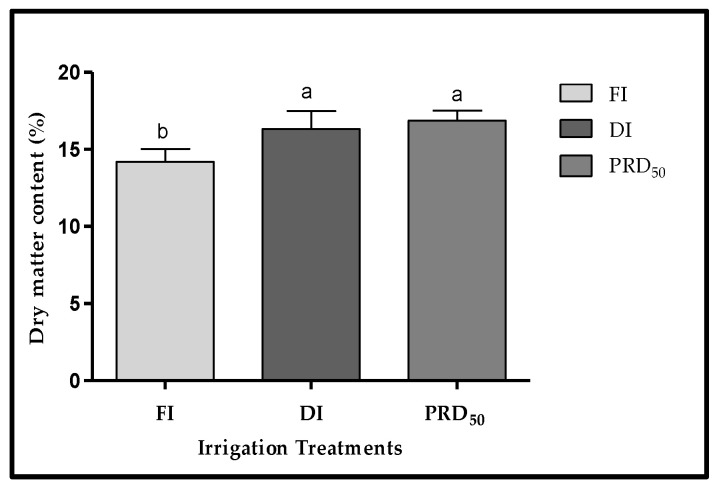
Fruit dry matter (±SD) of Flordastar peaches grown under FI, DI and PRD_50_ irrigation treatments, expressed in (%). Different letters refer to significant differences tested by Duncan’s multiple range test (*p* < 0.05).

**Figure 5 plants-11-01656-f005:**
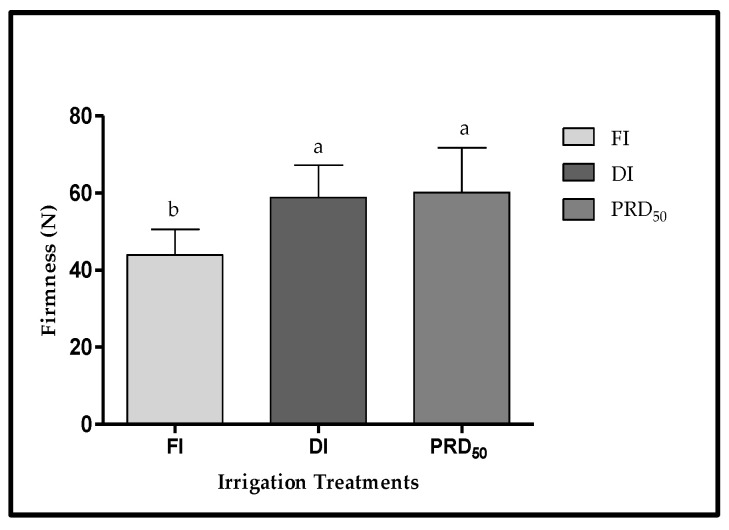
Fruit firmness (±SD) of Flordastar peaches grown under FI, DI and PRD_50_ irrigation treatments, expressed in (N). Different letters refer to significant differences tested by Duncan’s multiple range test (*p* < 0.05).

**Figure 6 plants-11-01656-f006:**
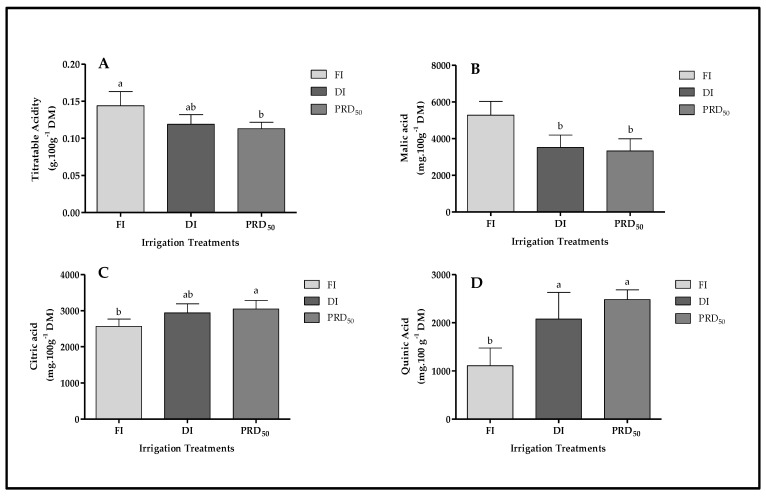
Titratable acidity (**A**) and organic acid content (malic acid (**B**), citric acid (**C**) and quinic acid (**D**)) in Flordastar peaches grown under FI, DI and PRD_50_ irrigation treatments, expressed in mg 100 g^−^^1^ DM. Data are mean values ± SD. Different letters refer to significant differences tested by Duncan’s multiple range test (*p* ≤ 0.05).

**Figure 7 plants-11-01656-f007:**
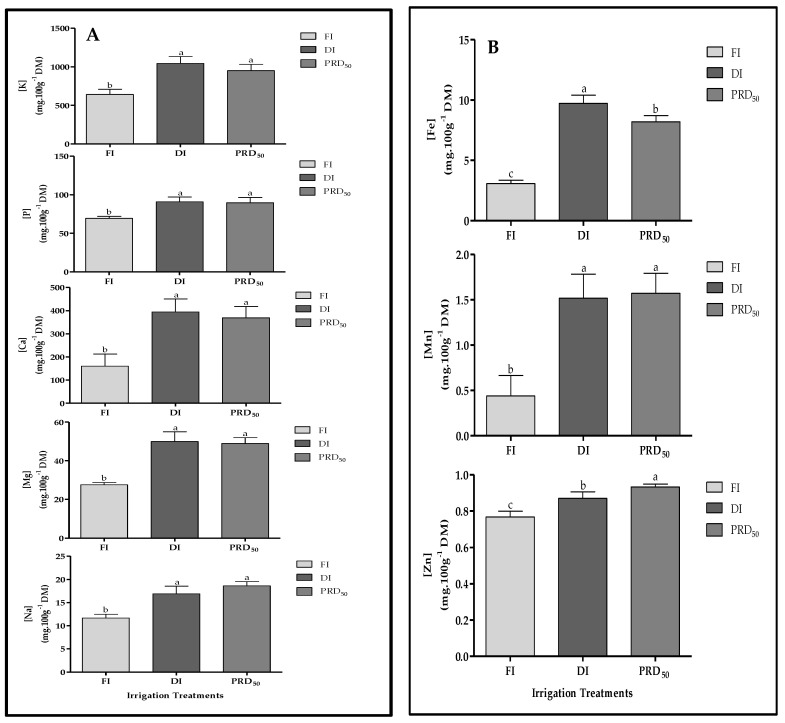
Macronutrient (**A**) and micronutrient (**B**) content in Flordastar peaches grown under FI, DI and PRD50 irrigation treatments, expressed in mg 100 g^−1^ DM. Data are means ± SD. Different letters refer to significant differences tested by Duncan’s multiple range test (*p* < 0.05).

**Figure 8 plants-11-01656-f008:**
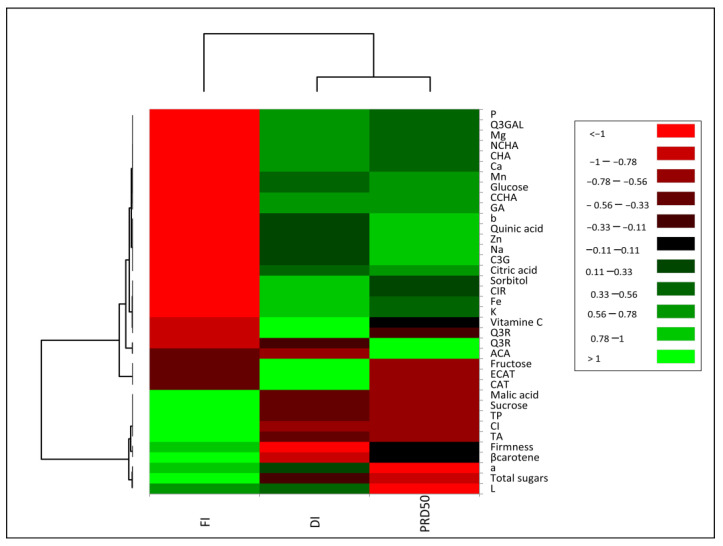
Heat map representation of Flordastar peach fruit quality traits grown under FI, DI and PRD_50_ irrigation treatments.

**Table 1 plants-11-01656-t001:** Applied irrigation water quantity (mm) by phenological stage and irrigation treatment of Flordastar peach tree.

Irrigation Treatment	Phenological Stage				Total
	Bud Break–Full Bloom	Full Bloom–Fruit Set	Fruit Set–Harvest	Post-Harvest	
**FI**	85.25	102.50	196.85	597.50	982.14
**DI**	42.63	51.25	98.42	298.76	491.07
**PRD_50_**	42.63	51.25	98.42	298.76	491.07

Values express water amount in mm.

**Table 2 plants-11-01656-t002:** Soluble-solid content (SSC) and sugar contents in Flordastar peach fruit samples harvested in 2016 subjected to different irrigation treatments (FI, DI and PRD_50_).

Irrigation Treatment	SSC	Sugars (g 100 g^−1^ DM)				Total Sugars (g 100 g^−1^ DM)
	(°Brix)	Sucrose	Fructose	Glucose	Sorbitol	
**FI**	12.3 ± 0.72 ^b^	54.28 ± 5.66 ^a^	7.40 ± 0.84 ^ab^	7.78 ± 0.74 ^b^	2.73 ± 0.66 ^a^	72.18 ± 3.63 ^a^
**DI**	14.3 ± 0.15 ^a^	46.04 ± 4.28 ^a^	8.55 ± 0.58 ^a^	9.93 ± 1.27 ^a^	3.42 ± 0.37 ^a^	67.94 ± 4.27 ^a^
**PRD_50_**	14.0 ± 0.45 ^a^	45.30 ± 3.65 ^a^	7.39 ± 0.26 ^b^	10.11 ± 0.63 ^a^	3.18 ± 0.70 ^a^	65.74 ± 4.32 ^a^
**Significance**	*	ns	*	*	ns	ns

Data are expressed in mean ± SD. In the same column, different superscript letters indicate significant differences tested by Duncan’s multiple range test (*p* < 0.05). The significance is noted by asterisks as follows: *—statistically significant differences at *p*-value ≤ 0.05; ns—not significant.

**Table 3 plants-11-01656-t003:** Total phenol content (mg GAE 100 g^−1^ DM) and individual phenolic compounds (mg 100 g^−1^ DM) in Flordastar peach fruits under different irrigation treatments (FI, DI and PRD_50_).

Irrigation Treatment	FI	DI	PRD_50_	Significance
**Phenolic acid**				
**GA**	0.14 ± 0.03 ^b^	0.28 ± 0.08 ^a^	0.28 ± 0.03 ^a^	*
**Hydrocinnamic acids**				
**CCHA**	85.42 ± 13.05 ^b^	153.49 ± 53.31 ^a^	154.82 ± 37.19 ^a^	*
**CHA**	0.26 ± 0.07 ^b^	0.67 ± 0.14 ^a^	0.62 ± 0.12 ^a^	*
**NCHA**	0.09 ± 0.1 ^b^	0.22 ± 0.04 ^a^	0.20 ± 0.03 ^a^	**
**Flavan-3-ols**				
**CAT**	3.63 ± 0.62 ^b^	5.15 ± 0.34 ^a^	3.34 ± 0.51 ^b^	*
**ECAT**	0.08 ± 0.006 ^b^	0.14 ± 0.130 ^a^	0.07 ± 0.026 ^b^	**
**Anthocyanin**				
**C3G**	0.17 ± 0.03 ^b^	0.32 ± 0.04 ^a^	0.37 ± 0.04 ^a^	**
**Flavonols**				
**Q3GAL**	1.98a ± 0.52 ^a^	3.64a ± 1.07 ^a^	3.52a ± 0.92 ^a^	ns
**Q3R**	0.70b ± 0.24 ^b^	2.25a ± 0.36 ^a^	1.32b ± 0.32 ^b^	**
**Flavones**				
**CIR**	0.44 ± 0.07 ^a^	0.57 ± 0.08 ^a^	0.53 ± 0.03 ^a^	ns
**ACA**	5.18 ± 0.55 ^a^	5.06 ± 0.31 ^a^	5.67 ± 0.53 ^a^	ns
**Other antioxidants**				
**Vitamin C**	42.82 ± 8.97 ^a^	53.89 ± 8.16 ^a^	48.26 ± 11.62 ^a^	ns
**β-Carotene**	0.71 ± 0.15 ^a^	0.21 ± 0.02 ^c^	0.43 ± 0.05 ^b^	**
**TP**	724.49 ± 69.49 ^a^	706.58 ± 49.49 ^a^	704.00 ± 22.94 ^a^	ns

Values are mean ± SD. In the same line, different superscript letters refer to significant differences tested by Duncan’s multiple range test (*p* < 0.05). The significance is noted by asterisks as follows: *—statistically significant differences at *p*-value < 0.05; **—statistically significant differences at *p*-value < 0.01; ns—not significant.

## Data Availability

Not applicable.

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
