# Peer review of "Improving Peach Fruit Quality Traits Using Deficit Irrigation Strategies in Southern Tunisia Arid Area"

_plants, 2022, doi:10.3390/plants11131656_

Round 1
Reviewer 1 Report
The manuscript entitled "Improving Early maturing Peach Fruits Quality traits using Partial Rootzone Drying and Deficit irrigation in Southern Tunisian Warm Area" by Τoumi et al. provided very useful data concerning the impact of deficit irrigation on the qualitative attributes of early maturing peaches, cultivated in the North part of Africa.
The text must be checked by a native English speaker since there are parts that need extensive editing.
The title is rather extended and some words could be excluded.
Within the text, the bibliography should be only presented by the use of numbers and not names (maybe this remained during the writing of the manuscript).
Within the text, be careful with the gaps between the words.
Line 161, provide the Abbreviation of GAE.
Line 171, _ symbol was used. Scan the text for such leftovers.
To all the provided results, provide a % of increase or decrease (accordingly) or provide a number of fold increase or decrease. For example in line 248, provide the fruit firmness % of increase under the deficit irrigation regime concerning Full irrigation (FI).
All figures must have the same font and size. Also, the letters that signify the statistical significance must be at an equal height above the bars. Pay attention to the way the bars are presented within each graph.
Author Response
"Please see the attachment"

Reviewer 2 Report
The presented manuscript describes the influence of various irrigation methods on selected parameters of peach fruit. Although the studied problem is of great practical importance, I have numerous comments on it.
The work was prepared in an extremely careless and sloppy manner. It is full of citation errors, empty or not closed parentheses, missing commas, periods, added spaces and even replaced words (e.g. simple instead of sample). This significantly hinders the reception of the work and shows the attitude of the authors to this publication. The authors use various abbreviations for the same parameters (for sustained deficit irrigation, the authors use both SDI and DI, and ETc translates as both crop evapotranspiration and cultural evapotranspiration), or they use abbreviations without explanation in text (what is WUE, MED or CH?). Authors mix up different tenses (e.g. sentence 225-226). What are 2016 decades (line 229)? A 10-day period is not a decade. Subheadings in Table 4 are written once in bold, once center-aligned, and right-aligned. I also have a note of Figure 7 and its description (line 341). FI did not increase firmness (line 247 and figure 4), so there is a mistake in the description and on the heat-map. The errors are even in the references (e.g. in which journal was publication 1 published?). I also have comments on the description of methods and results. The description on line 191-194 is incomprehensible. Has the tissue been fragmented in this solution? What were the incubation conditions? Similarly, the information in subchapter 2.4.3 is missing. If the method has been modified, these modifications should be described. The notation of the units also differs from the accepted standards. In addition, in line 190 the authors write that the results are converted to DM or FW, and then all presented results are converted only to DM.
In addition, authors should also describe fruit parameters such as relative water content (in order to discuss whether the dry matter content increase is actually due to metabolic changes rather than dehydration of the plant) and the size of the fruit. If the obtained fruit is small, the superiority of deficit irrigation is less important for the economy than the authors suggest.
To sum up, the work requires thorough revision and reading by the authors before submission.
Author Response
"Please see the attachment"

Reviewer 3 Report
The objective of this study was to evaluate the impact of three different irrigation strategies. on internal and external fruit quality of an early maturing peach cultivar in hot and dry areas of Tunisia irrigated with saline water.
This work is of interest and fits with the profile of the magazine.
The results underline the relevance of deficit irrigation and partial root irrigation strategies to improve the internal quality of the fruit due to the higher concentration of primary and secondary metabolites.
No principal component analysis results are shown in the manuscript. They have not been of interest?
This study allows us to suggest that some degree of water stress in early ripening peach trees seems to offer an effective technique to save water and improve the quality parameters of the product and, therefore, the economic performance of the farmer.
References:
In reference number 63, put the year in bold, to standardize the format.
Put the scientific name in italics in the references: 37, 40, 55, 60 and 72.
Author Response
"Please see the attachment"

Reviewer 4 Report
The paper,, Improving early maturing peach fruits quality traits using partial root zone drying and deficit irrigation in Southern Tunisian warm area,” is written correctly. The authors sufficiently bring the reader to the paper's subject in the introduction chapter. They have also defined the purpose of the research. However, they should emphasize the novelty of the experiment conducted and its impact on science more strongly. Please consider adding testable hypotheses to the research description. When you describe the methodology (or in the supplementary materials), please provide the exact values of the kc coefficient used for the calculations and its source. In my opinion, the results chapter is written correctly, and the authors insightfully did the discussion of the results. However, the conclusions should be reworded to emphasise more the novelty of the paper and to make the results obtained by the authors more visible.
Editing remarks:
Please do not put a full stop after the title.
In line 59, there is an unnecessary parenthesis.
In line 73, please remove the citation in the form of surname et al. year and leave only the number of the cited item—the same in lines 79, 136, 377, 391, 393, and 413.
Please also review the entire manuscript carefully and remove all double spaces.
Author Response
"Please see the attachment"

Round 2
Reviewer 2 Report
The authors significantly improved the manuscript. In my opinion, it can be published in this form.